# Analysis of Bio-Based Fatty Esters PCM's Thermal Properties and Investigation of Trends in Relation to Chemical Structures

Rebecca Ravotti, Oliver Fellmann, Nicolas Lardon, Ludger J. Fischer, Anastasia Stamatiou * and Jörg Worlitschek

Competence Centre Thermal Energy Storage (TES), Lucerne University of Applied Sciences and Arts, 6048 Horw, Switzerland; rebecca.ravotti@hslu.ch (R.R.); oliver.fellmann@hslu.ch (O.F.); nicolas.lardon@hslu.ch (N.L.); ludger.fischer@hslu.ch (L.J.F.); joerg.worlitschek@hslu.ch (J.W.)
* Correspondence: anastasia.stamatiou@hslu.ch; Tel.: +41-(0)41-349-3297

**Abstract:** As global energy demand increases while primary sources and fossil fuels' availability decrease, research has shifted its focus to thermal energy storage systems as alternative technologies able to cover for the mismatch between demand and supply. Among the different phase change materials available, esters possess particularly favorable properties with reported high enthalpies of fusion, low corrosivity, low toxicity, low supercooling, thermal and chemical stability as well as biodegradability and being derived from renewable feedstock. Despite such advantages, little to no data on the thermal behavior of esters is available due to low commercial availability. This study constitutes a continuation of previous works from the authors on the investigation of fatty esters as novel phase change materials. Here, methyl, pentyl and decyl esters of arachidic acid, and pentyl esters of myristic, palmitic, stearic and behenic acid are synthesized through Fischer esterification with high purities and their properties are studied. The chemical structures and purities are confirmed through Attenuated Total Reflectance Infrared Spectroscopy, Gas Chromatography coupled with Mass Spectroscopy and Nuclear Magnetic Resonance Spectroscopy, while the determination of the thermal properties is performed through Differential Scanning Calorimetry and Thermogravimetric Analysis. In conclusion, some correlations between the melting temperatures and the chemical structures are discovered, and the fatty esters are assessed based on their suitability as phase change materials for latent heat storage applications.

**Keywords:** latent heat storage (LHS); phase change material (PCM); esters; thermal energy storage (TES); fatty esters; decyl esters; methyl esters; pentyl esters

## 1. Introduction

The effective utilization of renewable energy sources has the potential to lower the rising global energy demand and to help establish a future based on sustainable use of energy. In particular, thermal energy storage is a key element in this prospect to adjust the imbalance between heat demand and supply, as well as to limit wastes and overproduction of heat [1–4]. Three different types of thermal storages exist and can be discerned based on the form in which heat is stored: Sensible (sensible heat storage, SHS), latent (latent heat storage, LHS), or chemical (chemical heat storage, CHS). The functioning principles of these energy storage technologies have already been extensively described in several sources in the literature [5–9], and thus will not be explained further hereby. The so-called phase change materials (PCM) allow storage of latent heat during reversible phase transitions. Latent heat storage presents the advantage over established sensible heat storage of providing higher energy

densities due to the charging phase occurring at a constant temperature while the storage material is melting. Additionally, they allow the usage of more compact unit sizes and an isothermal behavior during heat transfer processes [6]. The PCM selection plays a major role in the efficiency of an LHS system. In order to achieve this, PCMs need to possess several properties among which are congruent phase change, thermal stability and high enthalpy of phase change. Moreover, additional desired properties include non-corrosiveness, non-flammability, low to no degree of supercooling, low to no degree of toxicity, biodegradability and sustainability [3]. Esters are a class of organic materials that derive from the combination of a carboxylic acid with an alcohol. Similar to alcohols and carboxylic acids, esters are biodegradable and can be bio-based if derived from renewable feedstock. However, they are characterized by higher chemical stability and lower corrosivity compared to their precursors. Although esters have already been limitedly tested and their thermal properties investigated, they are still far from being established as PCM. There is a clear lack of experimental data in regards to esters' thermal behavior and properties due to their low commercial availability and high prices. Stamatiou et al. [10], Sari et al. [11] and Aydin et al. [12,13] investigated the thermal behavior of different types of both commercially available and self-synthesized esters, namely linear saturated esters, esters derived from stearic acid coupled with glycerol and other linear alcohols, and even-numbered fatty acid esters of myristyl alcohol. Ravotti et al. [14] performed a study on fatty esters derived from five different types of even-numbered fatty acids (Myristic Acid MY ($C_{14}$), Palmitic Acid PA ($C_{16}$), Stearic Acid SA ($C_{18}$), Behenic Acid BE ($C_{22}$)), coupled with alcohols of different chain lengths (methanol ($C_1$) and 1-decanol ($C_{10}$)). All esters showed promising thermal properties with supercooling below 5 °C, thermal stability over three heating-cooling cycles, and congruent phase change ranging from 15 to 45 °C, which supports their suitability as PCM for low to mid temperatures applications such as space heating [15–17]. In addition, some trends similar to what had been reported by Noël et al. [18] and Yang et al. [19] correlating the chemical structure to the thermal properties have emerged. Although studies on the correlation between chemical structure and thermal properties have been previously conducted and reported in the literature [20–22], in order to allow esters to develop as established PCM for a sustainable utilization of energy, further insight is needed. Understanding fully the relationship between the compounds' chemical structure and their properties could help to gain the knowledge necessary to individuate the best suitable esters to be used as PCM for different scopes and applications without necessarily requiring prior testing.

This work presents an extension of the investigation performed by Ravotti et al. [14] on linear fatty esters coupled with alcohols of different lengths. Esters of all fatty acids listed above coupled with 5-pentanol ($C_5$), as well as esters of Arachidic acid (shortened AR) ($C_{20}$) with methanol ($C_1$), 1-pentanol ($C_5$) and 1-decanol ($C_{10}$), were synthesized and their properties investigated. All IUPAC names, trivial names, chemical structures and shortenings from the esters produced are reported in Figure A1. Methyl arachidate is commercially available for elevated prices, while all pentyl esters and decyl arachidate are only limitedly purchasable. Little to no mention of their thermal behavior has been found in the literature to the best of the authors' knowledge.

## 2. Materials and Methods

Arachidic acid, methanol (MeOH), and 1-decanol were acquired from Sigma Aldrich (St. Louis, MO, USA) with high purities (≥99%) as synthesis precursors. 1-Pentanol was purchased by Roth GmbH (Karlsruhe, Germany) with purity ≥98%. Following the same synthesis procedure reported by Ravotti et al. [14], concentrated sulfuric acid ($H_2SO_4$) was used as the acid catalyst, sodium sulfate anhydrous ($Na_2SO_4$) as a water-absorbing agent for elimination of water and ethyl acetate (EtOAc) as the organic solvent for extraction. All these chemicals were purchased from Sigma Aldrich with purities ≥99%. Chloroform (GC quality, ≥99.9%) was used as a solvent for Gas Chromatography coupled with Mass Spectroscopy (GC-MS) analysis and was purchased from Sigma Aldrich. All chemical materials listed above were used without any prior purification.

*2.1. Synthesis*

The esters were synthesized and purified according to the synthesis procedure proposed and validated by Ravotti et al. [14] through Fischer esterification with $H_2SO_4$ as a catalyst. The reaction mechanism and the proposed method are reported extensively in the source mentioned above. Generally, similar to decyl esters, pentyl esters had to be crystallized in methanol once or twice with a ratio of 10:1 alcohol:ester at $-20\,°C$ until no residual acid and alcohol were observed in Nuclear Magnetic Resonance (NMR), GC-MS and Attenuated Total Reflectance Infrared Spectroscopy (ATR-IR). Analogous to what is stated in the aforementioned source, in order to ensure statistical relevance to the results obtained and to guarantee reproducibility, all syntheses described in the next chapters were repeated three times.

*2.2. Characterization*

2.2.1. Differential Scanning Calorimetry (DSC)

The analysis of melting and crystallization temperatures was conducted via DSC $823^e$ by METTLER TOLEDO (Columbus, OH, USA) between $-25$ and $100\,°C$ at both 2 and $10\,°C/min$ heating rates. The DSC was calibrated with indium standard before the measurements, and the uncertainty of the instrument was $\pm0.1$ K. The samples were measured under a constant flow of nitrogen at a rate of 100 mL/min, and sample masses were typically between 5 and 15 mg. In order to test the stability of the samples, the heating/cooling cycles were repeated three times for each heating rate with the same method for every sample, for a total of six cycles per sample. Moreover, as stated previously, all syntheses were performed three times to ensure reproducibility. Therefore, each ester type was measured three times with six cycles per sample. The values reported were calculated as follows: For each replicate of the same ester the average for all six cycles at 10 and $2\,°C/min$ for $T_m$, $T_c$ and $\Delta H$ was derived. Afterwards, the average of the three final values obtained for each repetition was retrieved and is reported in the next sections alongside the relative standard deviation. Therefore, all uncertainty values reported in the next sections are based on replicate measurements, thus denoting the accuracy of the results. The melting and crystallization peaks were calculated by the $STAR^e$ software through the tangent method, and the enthalpies were obtained by integration of the corresponding peak. The degree of supercooling was estimated considering the difference between the onset melting peak and the onset crystallization peak reported from the instrument.

2.2.2. Thermal Gravimetric Analysis (TGA)

The samples' thermal stability and degradation were analyzed by TGA on a $STAR^e$ 2 System by METTLER TOLEDO in the temperature range between 25 and $600\,°C$ with a heating rate of $10\,°C/min$ and sample masses between 15 and 20 mg. The uncertainty of the instrument's balance is reported to be $\pm0.1\,\mu g$. For each sample, a blank measurement of the empty crucible was performed with the same method as described above for correct baseline subtraction. The starting degradation temperature was typically defined as the earliest temperature for which mass losses $\geq 5\%$ were observed. Similarly, the end temperature degradation was marked as the earliest temperature for which the mass loss was $\geq 99\%$.

2.2.3. Attenuated Total Reflectance Infrared Spectroscopy (ATR-IR)

In order to gain structural information on the samples produced, as well as the degree of purity and the kinetics of the reaction, ATR-IR Cary 630 by Agilent Technologies (Santa Clara, CA, USA) in the wavenumber range between 4000 and $600\,cm^{-1}$ with $4\,cm^{-1}$ resolution was used. No prior sample preparation was needed, and a background spectrum was registered every 30 min with 32 scans. Afterwards, a few mg of the samples were simply deposited on the diamond tip in either solid or liquid form and the spectrum was registered with 32 scans.

### 2.2.4. Gas Chromatography Coupled with Mass Spectroscopy (GC-MS)

Gas chromatograms were measured on a Perkin Elmer (Waltham, MA, USA) GC Clarus 590 connected to a Perkin Elmer MS Clarus SQ 8 S with EI standard source, with 4.0 mm Glass inlet liner with deactivated Wool in split mode 1:50 on a Perkin Elmer Elite 530 m × 250 × 0.25 μm column and 1.0 mL/min flow rate. The oven program was 100 °C for 2 min, then 10 °C/min heating rate to 300 °C and afterwards hold for 5 min. The temperature of the Mass Selective Detector (MSD) transfer line was 250 °C. Mass spectra were measured with electron ionization (EI) at 70 eV and a source temperature of 200 °C. The instrument was scanned between $m/z$ 50 and 500 at a scan time of 0.3 s and inter-scan delay of 0.04 s. Perfluoroterbutylamine (PFTBA, Sigma Aldrich) served for tuning of the MS. The samples were diluted in Chloroform with a concentration of 0.1 mg/mL and the injection volume was 1 μL.

### 2.2.5. Nuclear Magnetic Resonance (NMR)

All [1]H NMR spectra were recorded using a Bruker (Billerica, MA, USA) 400 MHz ([1]H) spectrometer at room temperature (unless otherwise stated). Chemical shifts (δ-values) are reported in ppm, and spectra were calibrated relative to the residual proton chemical shifts of deuterated Chloroform (CDCl$_3$, δ = 7.26).

## 3. Results

Similarly to the previous work by Ravotti et al. [14], prior to any thermal analysis the esters synthesized were fully characterized and their structures and purities confirmed. As such, the samples were analyzed through ATR-IR, GC-MS and NMR.

### 3.1. ATR-IR

All ATR-IR spectra recorded showed a high degree of purity with sharp C=O and C-O-C stretching peaks from the esters arising in all samples around 1750 and 1150 cm$^{-1}$ respectively (Figure 1). The stretching of the C-H bonds in the aliphatic chains are visible at 2850 and 2920 cm$^{-1}$. As proved by Ravotti et al. [14], in the case of residual unreacted carboxylic acid or alcohol, side peaks would be visible at 1705 (sharp), 3000–2100 (broad) and 3600–3100 cm$^{-1}$ from the stretching of C=O and O-H in the carboxylic acid group, and from O-H in the unbound alcohol group respectively. Due to the lower sensitivity of the ATR-IR with liquids compared to solids, the peaks arising from PEMY and PEPA, which as will be showed in the next sections are in liquid form at room temperature, appear with lower intensity and slightly shifted compared to the other esters.

### 3.2. GC-MS

While the ATR-IR could prove the formation of the ester group and the purity of the samples produced, it could not directly allow to quantify the purities and to identify the structure of the ester analyzed. As such, GC-MS and NMR were used for this purpose. Table 1 shows the retention times retrieved and the main fragmentation peaks observed in GC-MS.

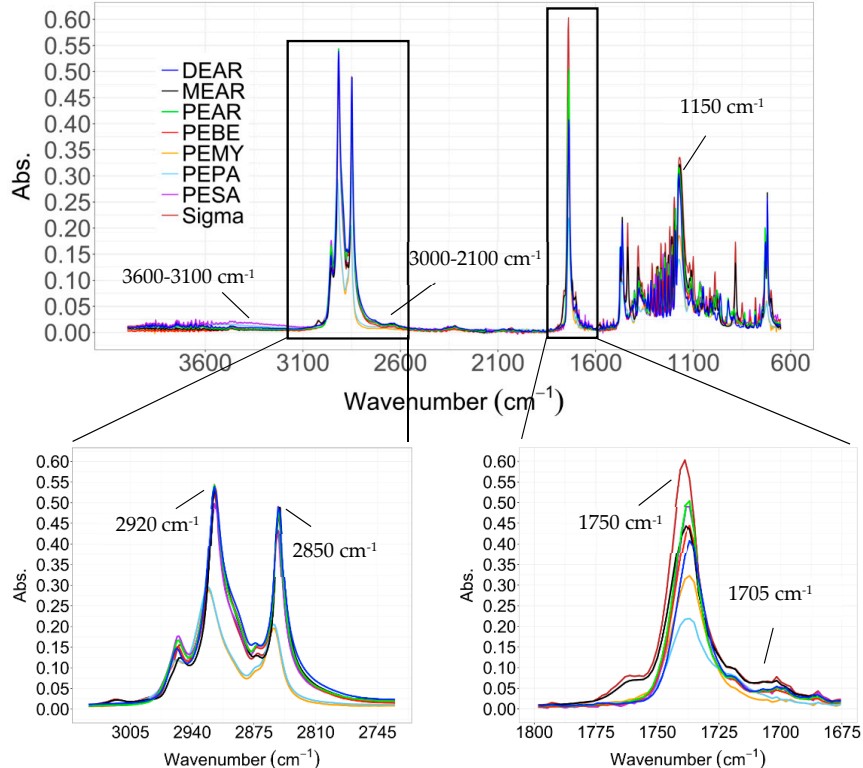

**Figure 1.** Attenuated total reflectance infrared spectroscopy (ATR-IR) of methyl, pentyl and decyl esters synthesized in this study compared to commercial methyl palmitate (indicated ad Sigma). The appearance of the peaks at 1750 and 1150 cm$^{-1}$ from the stretching of C=O and C-O-C in esters confirm the formation of the ester group. Impurities from unreacted alcohol or carboxylic acid would be visible at 3600–3100 and 3000–2100 cm$^{-1}$ from the stretching of free O-H in alcohols and carboxylic acids respectively, and at 1705 cm$^{-1}$ from the C=O stretching of carboxylic acids.

**Table 1.** Retention times in min derived from the gas chromatography (GC) chromatograms measured and fragmentation patterns reported from mass spectroscopy (MS) spectra with relative intensities in brackets. The most abundant fragment is assigned a value of 100% while all other fragments' intensities are calculated as a relative percentage of the main fragment.

| Compound | Retention Time GC, min | Fragmentation Peaks MS, *m/z* with Relative Intensities (%) |
|:---:|:---:|:---|
| PEMY | 16.53 | 298 (4), 255 (2), 229 (44), 211 (22), 199 (4), 185 (9), 171 (3), 157 (2), 143 (9), 129 (11), 115 (9), 97 (8), 83 (9), 70 (100), 57 (22) |
| PEPA | 18.26 | 326 (7), 284 (1), 257 (42), 239 (16), 227 (3), 213 (6), 199 (5), 185 (5), 171 (4), 157 (6), 143 (7), 129 (9), 115 (11), 97 (9), 83 (10), 70 (100), 57 (22) |
| PESA | 19.86 | 354 (7), 311 (1), 285 (30), 267 (10), 241 (7), 227 (1), 213 (1), 199 (4), 185 (5), 171 (2), 157 (1), 143 (7), 129 (11), 115 (10), 97 (8), 83 (11), 70 (100), 57 (28) |
| PEBE | 22.70 | 411 (12), 341 (29), 323 (7), 297 (4), 281 (1), 255 (2), 241 (3), 227 (1), 207 (8), 199 (4), 185 (5), 143 (8), 131 (9), 115 (10), 97 (10), 83 (10), 70 (100), 57 (30) |
| MEAR | 18.64 | 326 (19), 295 (5), 283 (14), 269 (3), 255 (4), 241 (7), 227 (7), 213 (4), 199 (7), 185 (8), 171 (5), 157 (4), 143 (22), 129 (9), 115 (5), 101 (8), 87 (71), 74 (100), 57 (19) |
| PEAR | 21.33 | 382 (14), 340 (1), 313 (34), 295 (12), 283 (1), 269 (8), 255 (1), 241 (2), 227 (2), 213 (4), 199 (6), 185 (4), 171 (3), 157 (2), 143 (6), 129 (10), 115 (11), 97 (10), 83 (11), 70 (100), 57 (33) |
| DEAR | 24.52 | 453 (8), 341 (1), 327 (1), 313 (50), 295 (4), 281 (6), 269 (5), 253 (4), 227 (1), 207 (23), 185 (6), 171 (1), 157 (1), 140 (57), 129 (7), 115 (3), 111 (35), 97 (34), 85 (36), 69 (40), 57 (100) |

Concerning GC analysis, no additional peaks beside the ester one were visible in the chromatogram, thus denoting the high purity of the compounds synthesized. Moreover, as expected, the retention times could be seen to increase for increasing carbon chain length due to higher apolarity, and consequently longer retention of the sample in the column. In MS spectra the molecular peak could be observed for all samples with low relative intensity for pentyl and decyl esters ($\leq$15%), and higher for MEAR (~20%). The loss of the alkyl alcohol carbon chain and formation of an enolate ion from the molecular ion was visible for pentyl and decyl esters with high abundances ($\geq$29%). This corresponded to losses of 70 $m/z$ and 140 $m/z$ for pentyl and decyl esters respectively.

In addition, all pentyl esters showed a peak deriving from the loss of 87 $m/z$ corresponding to the loss of 1-pentanol. In methyl esters, alcohol was lost as methanol by 31 $m/z$ thus forming an acylium ion, but was less visible (~5%) (Figure 2) [23].

**Figure 2.** Main fragments arising from the cleavage or breakage of the alcohol chain in the esters formed. Methyl esters lose the alcohol chain as methanol (31 $m/z$) thus forming an acylium ion. Pentyl and decyl esters undergo the loss of the alkyl chain (70 $m/z$ for pentyl and 140 $m/z$ for decyl esters) from the alcohol side forming an enolate ion.

During fragmentation in MS analysis, molecules containing a carbonyl group such as ketones, carboxylic acids and esters usually break through McLafferty rearrangement [24,25]. Such reaction sees the carbonyl group rearrange to form an enolate and an olefin by cleavage of a β-hydrogen. Thus, McLafferty fragments are characteristic of these molecules and usually visible with high intensities. In this case, the fragment formed from the esters' McLafferty rearrangement was visible at 57 $m/z$ with abundances between 20% and 30% for methyl and pentyl esters. Concerning decyl arachidate, this was the most abundant peak as expected; in fact, in longer alcohol-chained esters, Mclafferty's rearrangement constitutes the most prevalent phenomenon [24,25]. The presence of the McLafferty peak at 57 $m/z$ instead of 60 $m/z$ is suspected to be caused by further rearrangements in the enolate structure. Methyl arachidate gave rise to a second product to Mclafferty's rearrangement at 74 $m/z$ due to the presence of the methyl group (Figure 3).

**Figure 3.** Mechanism and products of McLafferty's rearrangement, typically occurring for esters in MS analysis.

Aside from the main fragments reported above, the typical fragmentation pattern spaced 14 $m/z$ units apart surging from the progressive fragmentation of the alkyl chain is visible in all samples. Thus, the structure and purity of the esters synthesized could be confirmed.

*3.3. NMR*

NMR analysis was performed alongside GC-MS to further confirm the purity and structures of the esters presented hereby. Table 2 reports the main peaks observed with relative coupling, integrals and interpretation. Generally, analogous to what was reported by Ravotti et al. [14] the main differences

which allowed to discern between methyl, pentyl and decyl esters could be observed in the peak of the hydrogens in $\alpha$ position to the ester group on the alcohol side. This appeared as a singlet at 3.69 ppm with normalized value 3 in the case of methyl esters due to the methyl hydrogens resonating together. On the other hand, it was visible as a triplet at 4.07–4.04 ppm with normalized value 2 in the case of pentyl and decyl esters due to the presence of neighboring hydrogens. In addition, the different normalized value of the multiplet at 1.35–1.26 ppm from the $-CH_2$ in the aliphatic chains resonating together due to similar environment allowed the identification of the chain length for each compound [23], thus confirming the chemical structure and the results previously obtained with the GC-MS. In the case of impurities, a peak at 3.66 ppm was observed corresponding to the $-CH_2$ in $\alpha$ to the alcohol group in a free unreacted alcohol molecule. However, the esters produced never presented impurities above 2.5%, confirming once again the high degree of purity observed previously through GC-MS and ATR-IR.

**Table 2.** List of $^1$H Nuclear Magnetic Resonance (NMR) peaks recorded in $CDCl_3$ from all samples produced, which allows the correct identification of compounds. Tetramethylsilane (TMS) was used as a reference and is assigned the value of 0 ppm. The peak position is reported in the peak section and the coupling in between brackets (s = singlet, t = triplet, m = multiplet). For each peak, the corresponding normalized value is reported in the integral column in the same order as the peak column, while the corresponding groups forming the signals are reported in the interpretation column.

| Compound | Chemical Structure | Peak (ppm/TMS) | Integral | Interpretation |
|---|---|---|---|---|
| **PEMY** ($C_5$-$C_{14}$) | $C_{19}H_{38}O_2$ | 4.07–4.04 (t), 2.31–2.27 (t), 1.64–1.55 (m), 1.35–1.26 (m), 0.92–0.86 (m) | 2, 2, 4, 24, 6 | $-CH_2$ in $\alpha$ (alcohol chain), $-CH_2$ in $\alpha$ (carbonyl), $-CH_2$ in $\beta$ (aliphatic chain), $-CH_2$ (aliphatic chains), $-CH_3$ (end aliphatic chains from both alcohol and acid sides) |
| **PEPA** ($C_5$-$C_{16}$) | $C_{21}H_{42}O_2$ | 4.07–4.04 (t), 2.31–2.27 (t), 1.64–1.59 (m), 1.35–1.25 (m), 0.91–0.86 (m) | 2, 2, 4, 28, 6 | $-CH_2$ in $\alpha$ (alcohol chain), $-CH_2$ in $\alpha$ (carbonyl), $-CH_2$ in $\beta$ (aliphatic chain), $-CH_2$ (aliphatic chains), $-CH_3$ (end aliphatic chains from both alcohol and acid sides) |
| **PESA** ($C_5$-$C_{18}$) | $C_{23}H_{46}O_2$ | 4.07–4.04 (t), 2.31–2.27 (t), 1.63–1.56 (m), 1.35–1.25 (m), 0.92–0.86 (m) | 2, 2, 4, 32, 6 | $-CH_2$ in $\alpha$ (alcohol chain), $-CH_2$ in $\alpha$ (carbonyl), $-CH_2$ in $\beta$ (aliphatic chain), $-CH_2$ (aliphatic chains), $-CH_3$ (end aliphatic chains from both alcohol and acid sides) |
| **PEBE** ($C_5$-$C_{22}$) | $C_{27}H_{54}O_2$ | 4.07–4.04 (t), 2.31–2.27 (t), 1.64–1.55 (m), 1.35–1.26 (m), 0.92–0.86 (m) | 2, 2, 4, 40, 6 | $-CH_2$ in $\alpha$ (alcohol chain), $-CH_2$ in $\alpha$ (carbonyl), $-CH_2$ in $\beta$ (aliphatic chain), $-CH_2$ (aliphatic chains), $-CH_3$ (end aliphatic chains from both alcohol and acid sides) |
| **MEAR** ($C_1$-$C_{20}$) | $C_{21}H_{42}O_2$ | 3.69 (s), 2.34–2.30 (t), 1.68–1.58 (m), 1.27 (m), 0.92–0.88 (t) | 3, 2, 2, 32, 3 | $-CH_3$ (alcohol chain), $-CH_2$ in $\alpha$ (carbonyl), $-CH_2$ in $\beta$ (aliphatic chain), $-CH_2$ (aliphatic chain), $-CH_3$ (end carboxylic chain) |
| **PEAR** ($C_5$-$C_{20}$) | $C_{25}H_{50}O_2$ | 4.07–4.04 (t), 2.31–2.27 (t), 1.64–1.56 (m), 1.35–1.26 (m), 0.92–0.86 (m) | 2, 2, 4, 36, 6 | $-CH_2$ in $\alpha$ (alcohol chain), $-CH_2$ in $\alpha$ (carbonyl), $-CH_2$ in $\beta$ (aliphatic chain), $-CH_2$ (aliphatic chains), $-CH_3$ (end aliphatic chains from both alcohol and acid sides) |
| **DEAR** ($C_{10}$-$C_{20}$) | $C_{30}H_{60}O_2$ | 4.07–4.04 (t), 2.31–2.27 (t), 1.64–1.55 (m), 1.25 (m), 0.92–0.86 (t) | 2, 2, 4, 46, 6 | $-CH_2$ in $\alpha$ (alcohol chain), $-CH_2$ in $\alpha$ (carbonyl), $-CH_2$ in $\beta$ (aliphatic chain), $-CH_2$ (aliphatic chains), $-CH_3$ (end aliphatic chains from both alcohol and acid sides) |

### 3.4. Thermal Properties

Following structural identification and purity quantification, the esters' thermal properties were analyzed through different techniques in order to evaluate their suitability as PCM and to individuate possible correlations with their chemical structures. As such, DSC was used to measure the phase change transition temperatures, the degree of supercooling, the phase change enthalpies, the cycling stability and the reliability of the results in respect of different heating rates. The degradation temperature intervals were assessed through TGA analysis.

Table 3 reports the averages of all the thermal properties measured with relative standard deviations. The esters produced are characterized by phase change temperatures in the range between 9 and 41 °C. In general, it could be observed that the pentyl esters registered lower melting temperatures than the ones of methyl and decyl esters derived from the same carboxylic acid. Despite the tendency of DSC to overestimate the degree of supercooling, all samples showed consistent supercooling values lower than 10 °C for all heating-cooling cycles. In addition, after six cycles with different heating rates at 2 and 10 °C/min, no deviations higher than 1.5 °C from the average onset crystallization and melting temperatures were observed, suggesting that the esters are thermally stable. However,

to confirm such statement, further stability tests with longer cycling times and higher frequencies should be performed in future works. All esters possessed enthalpies of fusion ΔH above 150 J/g, with standard deviations below 40 J/g for both crystallization and melting transitions during all cycles at both heating rates and for three repetitions. Only MEAR and DEAR presented values above 200 J/g, in accordance with what was shown by Ravotti et al. [14] for methyl and decyl esters having enthalpies of fusion ≥190 J/g. As such, if compared to methyl and decyl esters, similar to what was observed for the melting points, pentyl esters presented the lowest values of ΔH in the whole series. Concerning degradation temperatures, all methyl, pentyl and decyl esters underwent mass losses ≥5% for temperatures ≥140 °C and were fully degraded in the range between 200 and 300 °C, thus following the trend, according to which, the longer the carbon chain the higher the degradation temperature.

## 4. Discussion

As can be observed in Table 3, by ordering the esters by increasing carbon number no trend is revealed between the chemical structure and the phase change temperatures, as for example PEBE has a higher $T_m$ than DEAR but shorter carbon chain. Therefore, the esters have been ordered in Table 4 according to increasing phase change temperatures. In order to allow us to comprehensively compare the fatty esters and to gain a clear overview of possible trends, the esters synthesized and tested by Ravotti et al. [14] were added to the series in the table. By doing so, some trends correlating the chemical structure and the phase change temperatures can be observed. While, as expected, the melting point is raised by the addition of carbons to the aliphatic chain, it appears that the length of the carboxylic acid chain has a higher impact on the final $T_m$ than the length of the alcohol chain. For example, when considering MEPA, DEPA and MESA, the first two are derived from the same carboxylic acid (palmitic acid $C_{16}$) but different alcohols (methanol $C_1$ and 1-decanol $C_{10}$), while MESA is formed from the same alcohol as MEPA but with a longer carboxylic acid (stearic acid, $C_{18}$). In this case, the onset melting temperature increases from MEPA (26.25 °C) to DEPA (29.03 °C) by 2.78 °C due to the increased length of the alcohol chain that gives rise to a firmer crystal packing. However, when increasing the carboxylic acid chain length of two carbons, the temperature increases from MEPA (26.25 °C) to MESA (35.63 °C) of 9.38 °C. Such effect is visible throughout the whole series, but appears to be more marked the longer the carboxylic chain. For instance, despite deriving from a shorter alcohol, MEBE has a higher $T_m$ than DEBE of 3.12 °C. In the same way, from MEMY to DEMY the melting temperature increases by 10.01 °C, while from MEMY to MEPA by 11.10 °C. As such, it appears the alcohol chain length has a higher influence on the melting point of esters derived from shorter carboxylic acids than from longer ones. Nevertheless, it can be concluded that, in general, the carboxylic acid chain length has a stronger impact on the final $T_m$ than the alcohol chain, with such trend being more marked for esters formed from longer carboxylic acids. This could be explained by referring to a theory reported by Bunn et al. [26], according to which structures with the ester group at the end of the chain (such as methyl esters) possess higher melting points that those that see the ester bond in the middle of the molecule (decyl esters). This is caused by the reported flexibility of the O-C=O ester bond, which is able to rotate freely. A rotation in the middle of the molecule would cause vibrations in the whole chain far greater than at the end of the chain. Such vibrations would then disrupt the crystal lattice stability thus lowering the melting point. Therefore, two opposite effects are suspected to take place with the addition of longer chained alcohols: On one hand, the longer chain generates stronger intermolecular interactions and increases the rigidity of the structure, while on the other hand it shifts the position of the ester group from the end of the molecule to the middle thus causing stronger vibrations. Hence, this would explain the reason behind the lower impact of the alcohol chain on the final melting point compared to the carboxylic chain.

Another trend that emerges from Table 4 is the consistency of pentyl esters to show markedly lower phase change transitions than the corresponding methyl and decyl esters derived from the same carboxylic acid. This can be easily observed in Figure 4 below.

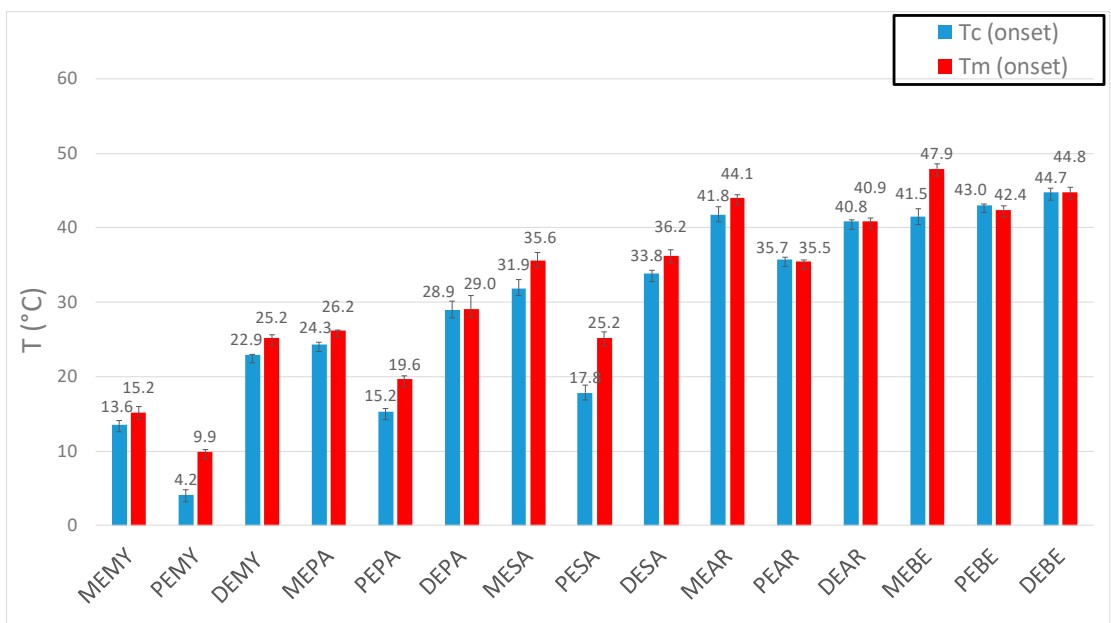

**Figure 4.** Barplot of all onset crystallization and melting temperatures of the esters reported in this study and in Ravotti et al. [14] work. Melting temperatures are indicated in red, while crystallization temperatures in blue. The esters are grouped according to both growing carboxylic acid and alcohol chain length. As such, MEMY, PEMY and DEMY are listed first since they are all derived from the shortest carboxylic acid in the sequence (myristic acid, $C_{14}$) and in order of growing alcohol chain length (methanol $C_1$, 1-pentanol $C_5$, 1-decanol $C_{10}$).

For example, PEPA (19.63 °C) has a lower melting point than MEPA (26.25 °C) by 6.62 °C, and PEBE (42.38 °C) is lower than MEBE (47.91 °C) by 5.53 °C. Such behavior could be attributed to an "odd-even" effect which had also been observed by Bunn et al. [26] in polymers. Here, the authors formulated the theory that odd-numbered chains formed less symmetrical structures due to their arrangement. In fact, at the end of odd $CH_2$ sequences, the bonds to the ester groups are inclined at an angle of 112°, whereas at the end of even sequences they are parallel. Methyl esters are technically odd-numbered as well, but thought not to follow this behavior due to a more "fixed" structure, since the methyl ends have merely one possible orientation. As such, the odd-effect is only visible for $CH_2$ chains longer than two carbon atoms. The odd-even effect does not affect only the phase change temperatures of the materials, but seems to influence the enthalpies of fusion as well. In fact, similarly to what was observed for melting temperatures, pentyl esters presented the lowest enthalpies of fusion of the whole series. This is possibly to be attributed to the odd-even effect described above and to the shifting of the ester group towards the middle of the molecule, which makes the crystal lattice subjected to stronger vibrations and disruptions. As such, these two behaviors together cause pentyl esters to possess lower symmetrical crystal packing.

The odd-even effect and the influence of the alcohol and carboxylic acid chain length on the final melting point can be observed in Figure 5.

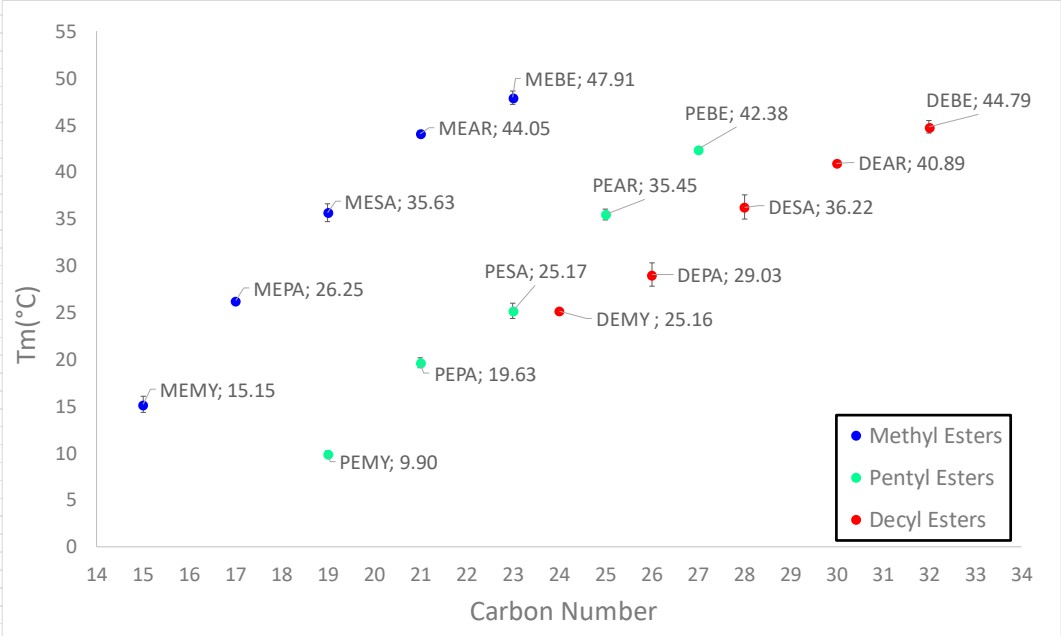

**Figure 5.** Scatter plot of the onset melting points of all esters reported in Table 4 for visual evaluation of the odd-even effect and the influence of the carboxylic acid chain length and the alcohol chain length on the phase change temperatures.

**Table 3.** Summary of thermal properties of the ester produced. Esters are ordered for increasing carbon number. The colors indicate the temperature gradient from the lowest values (blue) to the highest (red). A trend between degradation temperatures and molecular structure is observed, with progressive growth of the thermal stability together with carbon chain length. All ΔH (J/g) values registered were over 150 J/g, confirming the fatty acid esters' suitability as PCM.

| | Structure | Carbon Number | MW (g/mol) | Purity | $T_c$ (Onset, °C) | $T_m$ (Onset, °C) | Supercooling (°C) | ΔH (J/g) | ΔH (KJ/mol) | $T_{degradation}$ (Start, °C) | $T_{degradation}$ (End, °C) |
|---|---|---|---|---|---|---|---|---|---|---|---|
| PEMY ($C_5$-$C_{14}$) | $C_{19}H_{38}O_2$ | 19 | 298.50 | ≥95% | 4.17 ± 1.22 | 9.90 ± 0.51 | 5.73 | 177 ± 5 | 56.72 | 155 ± 35 | 273 ± 35 |
| PEPA ($C_5$-$C_{16}$) | $C_{21}H_{42}O_2$ | 21 | 326.57 | ≥95% | 15.24 ± 0.47 | 19.63 ± 0.48 | 4.39 | 187 ± 6 | 63.68 | 147 ± 7 | 277 ± 29 |
| MEAR ($C_1$-$C_{20}$) | $C_{21}H_{42}O_2$ | 21 | 326.57 | ≥95% | 41.75 ± 1.06 | 44.05 ± 0.35 | 2.30 | 213 ± 17 | 69.23 | 190 ± 14 | 330 ± 28 |
| PESA ($C_5$-$C_{18}$) | $C_{23}H_{46}O_2$ | 23 | 354.61 | ≥95% | 17.83 ± 0.98 | 25.17 ± 0.79 | 7.34 | 151 ± 39 | 59.22 | 163 ± 23 | 317 ± 15 |
| PEAR ($C_5$-$C_{20}$) | $C_{25}H_{50}O_2$ | 25 | 382.67 | ≥95% | 35.75 ± 0.28 | 35.45 ± 0.21 | 0.30 | 189 ± 31 | 70.79 | 213 ± 11 | 335 ± 7 |
| PEBE ($C_5$-$C_{22}$) | $C_{27}H_{54}O_2$ | 27 | 410.73 | ≥95% | 43.02 ± 0.20 | 42.38 ± 0.59 | 0.64 | 165 ± 19 | 67.77 | 220 ± 10 | 350 ± 10 |
| DEAR ($C_{10}$-$C_{20}$) | $C_{30}H_{60}O_2$ | 30 | 452.81 | ≥95% | 40.83 ± 0.17 | 40.89 ± 0.45 | 0.06 | 232 ± 24 | 105.96 | 240 ± 14 | 355 ± 7 |

**Table 4.** Fatty esters tested in this work as well as by Ravotti et al. [14] ordered according to increasing onset melting temperatures. As for Table 1, the colors indicate the temperature gradient from the lowest temperature values (blue) to the highest (red). Several coexisting trends between the chemical structures and the phase change temperatures are revealed.

| | Carbon Number | $T_c$ (Onset, °C) | $T_m$ (Onset, °C) |
|---|---|---|---|
| PEMY ($C_5$-$C_{14}$) | 19 | 4.17 ± 1.22 | 9.90 ± 0.51 |
| MEMY ($C_1$-$C_{14}$) | 15 | 13.58 ± 0.84 | 15.15 ± 1.50 |
| PEPA ($C_5$-$C_{16}$) | 21 | 15.24 ± 0.47 | 19.63 ± 0.48 |
| PESA ($C_5$-$C_{18}$) | 23 | 17.83 ± 0.98 | 25.17 ± 0.79 |
| DEMY ($C_{10}$-$C_{14}$) | 24 | 22.89 ± 0.16 | 25.16 ± 0.44 |
| MEPA ($C_1$-$C_{16}$) | 17 | 24.33 ± 0.24 | 26.25 ± 0.06 |
| DEPA ($C_{10}$-$C_{16}$) | 26 | 28.92 ± 1.24 | 29.03 ± 1.87 |
| PEAR ($C_5$-$C_{20}$) | 25 | 35.75 ± 0.28 | 35.45 ± 0.21 |
| MESA ($C_1$-$C_{18}$) | 19 | 31.86 ± 1.14 | 35.63 ± 0.97 |
| DESA ($C_{10}$-$C_{18}$) | 28 | 33.83 ± 1.29 | 36.22 ± 0.79 |
| DEAR ($C_{10}$-$C_{20}$) | 30 | 40.83 ± 0.17 | 40.89 ± 0.45 |
| PEBE ($C_5$-$C_{22}$) | 27 | 43.02 ± 0.20 | 42.38 ± 0.59 |
| MEAR ($C_1$-$C_{20}$) | 21 | 41.75 ± 1.06 | 44.05 ± 0.35 |
| DEBE ($C_{10}$-$C_{22}$) | 32 | 44.71 ± 0.63 | 44.79 ± 0.68 |
| MEBE ($C_1$-$C_{22}$) | 23 | 41.47 ± 1.03 | 47.91 ± 0.71 |

## 5. Conclusions and Outlook

To summarize, in this study pentyl esters (from 1-pentanol, $C_5$) of myristic $C_{14}$, palmitic $C_{16}$, stearic $C_{18}$, arachidic $C_{20}$ and behenic $C_{22}$ acid, alongside methyl and decyl arachidate (from methanol $C_1$ and 1-decanol $C_{10}$ with arachidic $C_{20}$ acid), were successfully synthesized through Fischer esterification in acidic catalytic conditions with purities over 95% and their properties analyzed. In order to prove the efficacy of the synthesis procedure and to confirm the formation of the desired compounds, the esters' chemical structures were identified through ATR-IR, GC-MS and NMR. Afterwards, the thermal properties were retrieved through DSC and TGA analysis. Methyl and decyl arachidate were found to have enthalpies of fusion above 200 J/g, while the pentyl esters showed lower values but still above 150 J/g. All esters presented good short-term stability with deviations in the phase change transition temperatures lower than 5%. Despite the tendency of DSC analysis to enhance the degree of supercooling perceived, the supercooling measured in the samples never exceeded 8 °C. Overall, the phase change temperatures of the esters measured were in the range between 10 and 48 °C. Upon comparison with the esters produced previously by Ravotti et al. [14], some trends correlating the chemical structures of the esters to the melting temperatures could be deduced. In particular, the aliphatic chain length of the carboxylic acid was found to have a higher impact on the increase of the melting temperature than the alcohol aliphatic chain. This is due to the rotational freedom of the ester bond, which, if shifted from the position at the end of the chain to the middle, creates stronger vibrations in the crystalline structure able to disrupt the lattice. In addition, an "odd-even" effect was discovered, with pentyl esters having constantly lower melting points in comparison to decyl esters. This is due to the fact that in even-numbered structures the bonds to the ester group are symmetrical, while in odd structures they are thought to be angled at 112°. Such an effect is only visible for $CH_2$ chains longer than two carbon atoms, and as such is not manifesting in methyl esters, which show the highest melting points of the whole series. The existence of trends correlating directly the chemical structure to the thermal properties is of significant importance since it could allow the development of predicting tools capable of calculating the chemical structure needed to achieve certain phase change temperatures for specific applications.

As a result of the promising results obtained, fatty esters appear to be interesting candidates as bio-based PCM for latent heat storage applications. However, pentyl esters might face bigger challenges compared to methyl and decyl esters regarding their implementation in real case applications due to lower enthalpies of fusion. Generally, given the temperature range retrieved, all esters presented could prove to be an interesting alternative to inorganic PCM or paraffins for low-mid temperatures application. One example constitutes the use of PCM embedded in textiles to create "smart" sportswear to reduce sweat production and thus hot and cold peaks in the human body. Such application usually requires PCM with phase change transitions between 28 and 33 °C. Another possible field is represented by space cooling and heating applications, which necessitate of materials with phase change transitions generally below 23 °C for space cooling and above 25 °C for space heating. PCM with higher phase transition, such as 40 to 50 °C can be used for cooling of electronic devices and machining tools [27,28]. According to the European Commission [4], water heating accounts for 79% of total final household energy use, while cooling shares a lower but increasing percentage due to recent climate change. For both, 84% of such energy is still derived from fossil fuels. As such, the introduction of bio-based PCM from renewable feedstock could have a major impact on the energy consumption turnaround and the reduction of fossil fuels usage scenario.

Given the lack of experimental thermal data on esters, future research on other ester classes (e.g., lactones, diesters, triglycerides, aromatic esters) is planned in order to collect further information about the thermal behavior and possible trends, and to individuate the best suitable candidates to be used as PCM for different scopes and applications.

**Author Contributions:** All authors listed conceived and designed the project and contributed to experiments' development and in reviewing the paper; A.S., J.W. and L.J.F. acquired the funding; R.R. and N.L. performed the

experiments; R.R., N.L., O.F. and A.S. analyzed the data; O.F. contributed reagents, materials and analysis tools; R.R. wrote the paper.

**Funding:** This research was funded by the Swiss National Science Foundation (SNSF, project number PZENP2_173636).

**Acknowledgments:** This work was developed within the framework of the project DENSE "Direct-contact ENergy StoragE" funded by the Swiss National Science Foundation (SNSF, project number PZENP2_173636) with the support of the Swiss Competence Center for Energy Research Storage of Heat and Electricity (SCCER). The authors wish to thank the University of Zürich (UZH, Zürich, Switzerland) for the NMR measurements conducted.

**Conflicts of Interest:** The authors declare no conflict of interest.

## Abbreviations

| | |
|---|---|
| AR | Arachidic Acid |
| ATR-IR | Attenuated Total Reflectance InfraRed Spectroscopy |
| BE | Behenic Acid |
| $CDCl_3$ | Deuterated Chloroform |
| CHS | Chemical Heat Storage |
| DEAR | Decyl Arachidate ($C_{10}$-$C_{20}$) |
| DEBE | Decyl Behenate ($C_{10}$-$C_{22}$) |
| DEMY | Decyl Myristate ($C_{10}$-$C_{14}$) |
| DEPA | Decyl Palmitate ($C_{10}$-$C_{16}$) |
| DESA | Decyl Stearate ($C_{10}$-$C_{18}$) |
| DSC | Differential Scanning Calorimetry |
| EtOAc | Ethyl Acetate |
| GC-MS | Gas-Chromatography coupled with Mass Spectrometry |
| $H_2SO_4$ | Sulfuric Acid |
| LHS | Latent Heat Storage |
| MEAR | Methyl Arachidate ($C_1$-$C_{20}$) |
| MEBE | Methyl Behenate ($C_1$-$C_{22}$) |
| MEMY | Methyl Myristate ($C_1$-$C_{14}$) |
| MeOH | Methanol |
| MEPA | Methyl Palmitate ($C_1$-$C_{16}$) |
| MESA | Methyl Stearate ($C_1$-$C_{18}$) |
| MSD | Mass Selective Detector |
| MY | Myristic Acid |
| $Na_2SO_4$ | Sodium Sulfate |
| NMR | Nuclear Magnetic Resonance |
| PA | Palmitic Acid |
| PCM | Phase Change Material |
| PEAR | Pentyl Arachidate ($C_5$-$C_{20}$) |
| PEBE | Pentyl Behenate ($C_5$-$C_{22}$) |
| PEMY | Pentyl Myristate ($C_5$-$C_{14}$) |
| PEPA | Pentyl Palmitate ($C_5$-$C_{16}$) |
| PESA | Pentyl Stearate ($C_5$-$C_{18}$) |
| PFTBA | Perfluorotributylamine |
| SA | Stearic Acid |
| SHS | Sensible Heat Storage |
| $T_c$ | Crystallization Temperature |
| TCS | Thermochemical Storage Material |
| TES | Thermal Energy Storage |
| TGA | Thermogravimetric Analysis |
| $T_m$ | Melting Temperature |
| TMS | Tetramethylsilane |
| $\Delta H$ | Enthalpy of fusion |

## Appendix A

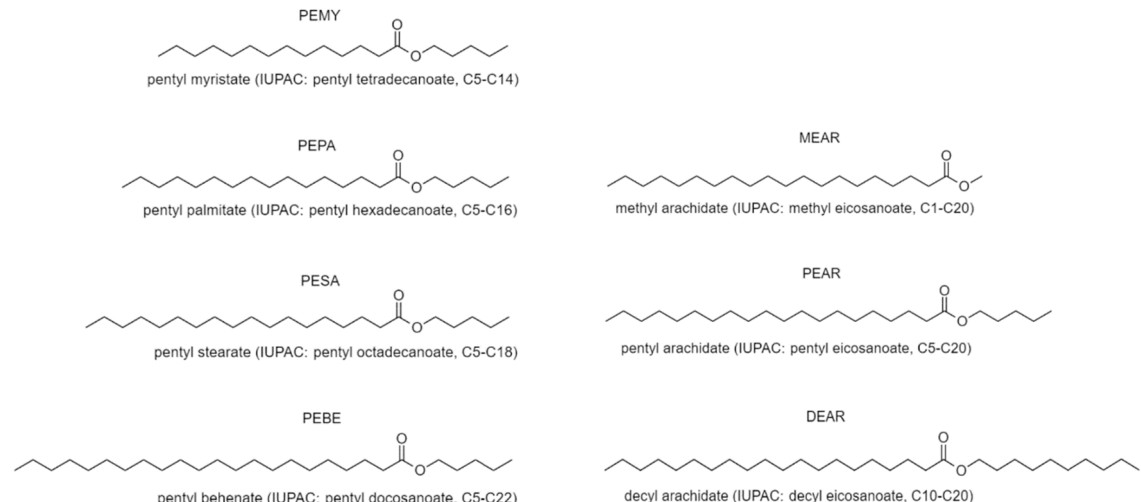

**Figure A1.** Structures and trivial names of esters produced, with IUPAC nomenclature and carbon length shortening in between brackets. The carbon length shortening indicates first the number of carbons in the alcohol, and subsequently the number of carbons in the carboxylic acid chain. For example, pentyl arachidate derives from 1-pentanol ($C_5$) and arachidic acid ($C_{20}$), therefore is indicated as $C_5$-$C_{20}$.

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
