# Peer review of "Analysis of Bio-Based Fatty Esters PCM’s Thermal Properties and Investigation of Trends in Relation to Chemical Structures"

_applsci, doi:10.3390/app9020225_

Round 1
Reviewer 1 Report
It is a very timely and interesting paper. Just some minor considerations proposed to the authors:
i) minor corrections on the English language
ii) just present in a more clear manner your error analysis
Author Response
Dear Reviewer,
Thank you for your time and your insightful suggestions.
Please notice that the following corrections have been implemented according to your comments:
i) the english language has been proofread and edited;
ii) A paragraph has been added at line 113-118 explaining in details how the values and relative errors were calculated.
Please mind that the corrections can be followed in details using the word changes tracker.
Best Regards,
The Authors
Reviewer 2 Report
In the paper titled: “Analysis of bio-based Fatty Esters PCM’s Thermal Properties and Investigation of trends in relation to Chemical Structures” is discussed an extension of the investigation performed by Ravotti et al. on linear fatty esters coupled with alcohols of different lengths. The paper is well written and organised.
Some minor revisions are due:
1) The style adopted in the text and in the list of references is not uniform and not correct for the present journal.
2) The authors could consider this paper: Sepe, R., Armentani, E., Pozzi, A. Development and stress behaviour of an innovative refrigerated container with PCM for fresh and frozen goods (2015) Multidiscipline Modeling in Materials and Structures, 11 (2), pp. 202-215 in order to improve the introduction (e.g. at line 40).
3) Line 114: insert “/” between °C and min.
4) Improve the quality and resolution of Figure A1.
5) In the text there are some typos, please check the text.
6) Moderate proof English is due.
Author Response
Dear Reviewer,
Thank you for your time and your insightful suggestions.
Please notice that the following corrections have been implemented according to your comments:
1) The text style is now uniform throughout the whole document;
2) The paper suggested has been carefully read; given its valuable information, it has now been implemented in the text at line 40;
3) The "/" has been inserted at line 114;
4) Figure A1 has been changed to a higher resolution one;
5) Several typos have been found and corrected accordingly. The authors sincerely apologize for the inconvenient;
6) The english language has been proofread and edited.
Please mind that the corrections can be followed in details using the word changes tracker.
Best Regards,
The Authors